# The Human Coronary Collateral Circulation, Its Extracardiac Anastomoses and Their Therapeutic Promotion

**DOI:** 10.3390/ijms20153726

**Published:** 2019-07-30

**Authors:** Bigler Marius Reto, Christian Seiler

**Affiliations:** Department of Cardiology, Inselspital, Bern University Hospital, University of Bern, 3010 Bern, Switzerland

**Keywords:** human coronary collateral circulation, extracardiac anastomoses, collateral flow index, collateral artery growth in man, permanent internal mammary artery occlusion

## Abstract

Cardiovascular disease remains the leading global cause of death, and the number of patients with coronary artery disease (CAD) and exhausted therapeutic options (i.e., percutaneous coronary intervention (PCI), coronary artery bypass grafting (CABG) and medical treatment) is on the rise. Therefore, the evaluation of new therapeutic approaches to offer an alternative treatment strategy for these patients is necessary. A promising research field is the promotion of the coronary collateral circulation, an arterio-arterial network able to prevent or reduce myocardial ischemia in CAD. This review summarizes the basic principles of the human coronary collateral circulation, its extracardiac anastomoses as well as the different therapeutic approaches, especially that of stimulating the extracardiac collateral circulation via permanent occlusion of the internal mammary arteries.

## 1. Introduction

According to the American Heart Association, “cardiovascular disease is the leading global cause of death”, accounting for more than 17.6 million deaths in 2016, a number that is expected to grow to more than 23.6 million by 2030 [1] In the event of acute coronary syndrome, percutaneous coronary intervention (PCI) has been shown to be beneficial on outcome [2]. The beneficial effect of PCI on the course of chronic stable coronary artery disease (CAD) has, so far, not been proven yet [3]. A recently published randomized controlled trial among patients with stable, single-vessel CAD, the so called ORBITA trial (e.g., Objective Randomised Blinded Investigation With Optimal Medical Therapy of Angioplasty in Stable Angina) [4], found that PCI of the stenotic lesion did not prolong exercise time by more than the effect of a sham procedure during the short observation period of six weeks. The new aspect of the ORBITA trial was a methodological one, that is, the use of a sham control group of patients undergoing the invasive coronary procedure, but not the actual PCI. The importance of a sham control group in interventional procedures is pivotal, especially in a population with a high level of suffering [5,6]. After all, it is known that the placebo effect can cause significant clinical improvements (e.g., an increased exercise duration of >90 s [7]). Coronary artery bypass grafting (CABG), on the other hand, has been found superior to PCI with respect to all-cause or cardiovascular mortality [8].

The number of patients with incomplete revascularization as well as so-called “no-option”-patients (i.e., patients without options for PCI or CABG still suffering from symptoms of CAD despite optimal medical therapy) is on the rise. It is estimated, that 30,000–50,000 new patients are affected in continental Europe per year [9] and Williams et al. reported a prevalence of 25.8% of incomplete revascularization in patients with CAD [10]. Apart from the limited quality of life, these patients also have a higher mortality at three years than patients with complete revascularization [10].

Accordingly, new therapeutic approaches are required. Because of the known survival benefit of patients with a well-developed coronary collateral circulation [11,12], interventions aiming at the promotion of coronary collaterals are a promising strategy. Coronary collaterals represent pre-existing inter-arterial anastomoses and as such are the natural counterpart of surgically created bypasses. To this end, biochemical (e.g., intracoronary vascular-endothelial growth factor or intravenous granulocyte-macrophage colony-stimulating factor) as well as biophysical (e.g., external counterpulsation) approaches have been evaluated for the promotion of those collaterals.

The aim of this review is to describe basic principles of the coronary collateral circulation, its extracardiac anastomoses as well as different therapeutic approaches, especially that of stimulating extracardiac coronary supply via permanent occlusion of the internal mammary arteries.

## 2. Basic Principles of the Human Coronary Collateral Circulation

### 2.1. Coronary Collateral Circulation

The development of the cardiovascular system during embryogenesis occurs by vasculogenesis, a process defined as “the de novo formation of blood vessels from endothelial precursor cells” [13]. Directed by the concentration of local messenger substance, endothelial precursor cells sprout out and start forming a dense vascular network with multiple anastomoses. The density of this network is at its peak in neonates and declines subsequently by physiological regression, a process called pruning [14,15,16].

Nevertheless, it has been hypothesized early on and tested that the coronary anastomoses of the neonate do not vanish completely but some collaterals rather recede in calibre. This concept has been decisively advanced by the findings of the Scottish pathologist W.F. Fulton, who found “numerous anastomoses in all normal hearts” by using a vascular overlay detecting technique with radiographic contrast medium containing uniform particles sized 0.5–2.0 µm to visualize even small arteries [17].

Interestingly, with changing vascular pressure- and resistance conditions, it is possible to recruit these receded arterial anastomoses. This process is often seen during the course of CAD with development of a pressure gradient across a stenotic lesion, which itself induces augmented flow in preformed arterial anastomoses and finally, structural augmentation of these collateral arteries (arteriogenesis). Accordingly, the prevalence of functional coronary anastomoses depends on the presence of CAD and is highest in chronic total coronary occlusions [16].

Coronary collaterals in patients without coronary atherosclerosis range in calibre between 10–200 µm; collateral arteries of patients with CAD are approximately four times bigger (100–800 µm) [17]. This observation is in accordance with an experimental rabbit model, where occlusion of the femoral artery increased the lumen diameter of pre-existent arterioles four- to fivefold [18]. “At the same time, the growth in structural size goes along with a decreasing number of collateral arteries, a process called pruning. Pathophysiologically and in the sense of the Hagen Poiseuille law, pruning may be interpreted as a way of effectively reducing vascular resistance to collateral flow” [13].

### 2.2. Extracardiac Coronary Supply

Apart from inter-coronary arterial anastomoses, the human coronary arterial circulation is supplied by several extracardiac anastomoses, also called the non-coronary collateral myocardial blood flow (NCCMBF) [19]. Hence, the heart receives additional blood from the arteries of surrounding structures [20,21,22,23,24]. Most of the extracardiac anastomoses originate from arteries, which supply the pericardium [21] and these arteries are typically located at the sites of pericardial reflections (e.g., the entry of the caval veins or the exit of the great arteries) [22]. Thus, a well-known extracardiac anastomosis connects the right internal mammary artery (IMA, also called internal thoracic artery) to the right coronary artery via the pericardiacophrenic branch and the sinus node artery [25] (Figure 1). This extracardiac coronary supply can also develop after coronary bypass surgery as shown exemplary in Figure 2 [22].

Most commonly, NCCMBF originates from the bronchial or the internal mammary arteries [22]. Bjork et al. showed a prevalence for bronchial-coronary-anastomoses of more than 20% by reviewing 200 coronary angiographies [26]. According to this observation, most of the anastomoses connect to the left circumflex artery (LCX) and demonstrate poor blood flow. However, blood flow within an anastomosis between two arterial beds depends on the respective vascular resistances. Thus, a constant decrease of vascular resistance in one arterial bed causes an increased blood flow to it with associated arteriogenesis. Consequently and depending on the underlying pathology, bronchial-to-coronary (e.g., in the case of a chronic occluded coronary artery [27]) as well as coronary-to-bronchial anastomoses (e.g., during chronic pulmonary diseases [28]) have been described.

Additional evidence for extracardiac anastomoses comes from the work of Hudson et al., who, by injecting ink into the coronary arteries, demonstrated anastomoses with anterior mediastinal, phrenic and intercostal arteries as well as with esophageal arterial branches of the aorta [21].

NCCMBF has also been increasingly recognized by cardiac surgeons as they discovered that anastomotic blood flow can dilute, and thus, be a potential hazard to cardioplegia [23]. To quantify this phenomenon, several studies have been conducted with reported values of anastomotic perfusion ranging between 3.4 to 14 mL/100 g/min [29,30] during cardiopulmonary bypass with cross-clamping of the aorta.

### 2.3. Quantitative Evaluation of the Coronary Collateral Circulation

The first in vivo functional coronary collateral measurements were conducted in the 1970s, showing a direct relation between “angiographic appearance and functional performance of coronary collaterals during bypass surgery” [31]. Rentrop et al. proposed a transluminal coronary angioplasty approach, which divided the appearance of coronary collaterals in four groups (0 = no collateral filling from the contralateral vessel to 3 = “complete filling of the epicardial segment of the artery”) [32]. Unfortunately, the method is only qualitative and evaluation of extracardiac collaterals is not feasible.

Thereafter a method for quantitative coronary collateral function assessment based on coronary occlusive pressure measurements was introduced. The so called collateral flow index (CFI) [33,34] “is the ratio between mean coronary occlusive and aortic pressure both subtracted by central venous pressure as obtained during a 1-min proximal coronary balloon occlusion” [33] (Figure 3). The method is accepted as the reference method for functional collateral assessment in patients with chronic stable CAD [35,36]. In terms of sufficient collateral blood supply, it has been demonstrated that a CFI of >0.20–0.25 is related to absent signs of ischemia on the intracoronary electrocardiogram (i.c.ECG) during this 1-min coronary artery balloon occlusion [37,38].

CFI has also been determined in patients with angiographically normal coronary arteries, revealing functional collateral arteries “to the extent, that one fifth to one quarter of them (i.e., the patients without coronary stenoses) do not show signs of myocardial ischemia during the brief vascular occlusions” [39]. Those findings of functional sufficient collaterals even in the absence of CAD support the above mentioned pathoanatomic observations [17], that coronary anastomoses calibre remain functional to a considerable degree.

## 3. Angiogenesis and Arteriogenesis

To understand the different therapeutic approaches for promoting the coronary collateral circulation, it is crucial to differentiate between two basic physiologic principles, that is, angiogenesis and arteriogenesis.

### 3.1. Angiogenesis

The formation of capillaries from pre-existing vessels to expand the microvascular system by increasing the capillary density is called angiogenesis. Driven by several growth factors such as hypoxia-inducible factor 1α, vascular endothelial growth factor (VEGF) [40] and inflammatory mediators as well as inflammatory cells (mainly monocytes [41]), a local milieu is formed [42] which promotes the proliferation and migration of endothelial cells, pericytes and smooth muscle cells. Thereby, “the amplification of the vascular network occurs within a short time due to either abluminal outgrowth (sprouting) or intraluminal division (intussusceptive growth) of capillaries” [43]. In contrast to arteriogenesis, angiogenesis is mostly driven by metabolic demands (i.e., hypoxemia) [44]

### 3.2. Arteriogenesis

“Although capillary sprouting may deliver some relief to the underperfused territory, only true collateral arteries are principally capable of providing large enough amounts of blood flow to the ischemic area at risk for necrosis or loss of function.” [41] Hence, arteriogenesis is the process of outward remodelling [44] (i.e., growth in diameter and length) of pre-existing anastomoses [45], resulting in an increased flow capacity of the artery.

Fluid shear stress, “the product of spatial flow velocity changes during the cardiac cycle and blood viscosity” [46] “is the primary and strongest arteriogenic stimulus” [47]. It leads to the expression of nitric oxide (NO), VEGF and monocyte chemoattractant protein-1 (MCP-1), resulting in the attraction and activation of monocytes [41,44,48,49,50,51]. Those inflammatory cells conduct the process of arteriogenesis with induction of cell proliferation as well as preparation of the extracellular matrix to enable cell migration [48].

Arteriogenesis is a common phenomenon that interventional cardiologists encounter on a daily basis as it appears (e.g., in the course of hypertensive heart disease with concentric left ventricular hypertrophy and augmented myocardial mass). Due to the direct and curvilinear relationship between myocardial mass and coronary arterial cross-sectional area [52], structural remodelling (i.e., arteriogenesis of the epicardial coronary arteries) occurs, resulting in large vascular calibres and, because of undirected growth, also affecting vascular length. Thus, this leads to the typical corkscrew pattern that is seen in this condition (Figure 4A).

Importantly and in contrast to the previously outlined process of angiogenesis, myocardial ischemia is unrelated to this process [53,54]. Arteriogenesis depends solely on physical pressure gradients across pre-formed anastomoses between different arterial territories with consequent augmentation of endothelial fluid shear stress [47,53,55]. Figure 4B illustrates this concept: After myocardial biopsy with perforation of the LAD and consecutive drainage into the (low resistance) right ventricular cavity, blood flow in the LAD increased due to the abrupt decrease in “vascular” resistance. As a consequence, abundant growth of the LAD, both in cross-sectional area and length, could be observed [56].

## 4. Therapeutic Promotion of the Coronary Collateral Circulation

The following chapter summarizes the most promising therapeutic approaches of coronary collateral promotion divided according to the basic concept of biochemical or biophysical methods.

### 4.1. Biochemical Concepts

In general, biochemical concepts are “prone to potentially harmful effects, since arteriogenesis shares many common mechanisms with inflammatory diseases, such as atherosclerosis” [44]. Accordingly, Epstein et al. coined (biochemical) collateral promotion a “Janus Phenomenon”, that is, “whatever intervention enhances collaterals increases atherogenesis and vice versa” [57].

With the rapid development of angiogenic growth factors and the growing understanding of their mechanisms of action, multiple trials testing collateral growth promotion have been initiated. Because of the known pivotal role of monocytes in orchestrating the different processes of angio- and arteriogenesis [50], most of the projects focused on the activation or the recruitment of this cell line. Growth factors most extensively studied have been granulocyte-macrophage colony-stimulating factor (GM-CSF) [58,59,60,61,62], granulocyte colony-stimulating factor (G-CSF) [63,64,65,66] or monocyte chemoattractant protein-1 (MCP-1) [41]. Besides, also different fibroblast growth factors (FGF) [67,68,69] and VEGF [70] have been clinically tested. Altogether, this study showed that angiogenesis is less efficient than arteriogenesis [71] for promoting bulk collateral blood flow, since it only promotes microvascular density. Consequently, clinical trials evaluating the effect of angiogenetic factors such as FGF or VEGF have failed to demonstrate a therapeutic effect that exceeds the effect of placebo treatment [67,68,70].

Colony-stimulating factors, on the other hand, have been found to promote the formation of large interconnecting arterioles (arteriogenesis), which are required for the salvage of myocardium in the presence of occlusive CAD [58]. Buschmann et al. [61] found that a continuous infusion of GM-CSF into the stump of the acutely occluded femoral artery of rabbits enhanced blood flow to the hind limb five-fold. The mechanism of action in that study has been found to be the prolonged survival of monocytes, “known to play a decisive role in arteriogenesis” [61]. In two small but randomized and placebo-controlled clinical trials with 35 patients in total, GM-CSF has been shown to be efficacious in a short-term subcutaneous administration protocol of two weeks [58,59]. Both studies have demonstrated a significant increase in CFI (from 0.116 to 0.159; *p* = 0.028 respectively from 0.21 to 0.31; *p* < 0.05). Of note, this beneficial effect of GM-CSF in the promotion of coronary collateral growth could not be transferred to the clinical setting of peripheral vascular disease, where it failed to improve the walking time [60]. Further, one of the clinical trials using GM-CSF for arteriogenesis had to be stopped prematurely for safety concerns in the context of two patients with acute coronary syndrome in the treatment group [59].

G-CSF has been reported in meta-analyses to be safe in terms of major adverse cardiovascular events (cardiovascular death, recurrent myocardial infarction and in-stent restenosis) and toleration of the treatment injections [72,73,74]. These findings and promising animal test results [75] have led to a randomized, placebo-controlled clinical trial in humans, in which subcutaneous G-CSF was shown to increase CFI from 0.121 to 0.166 (*p* < 0.0001) when administered every other day for two weeks [63]. Despite the above meta-analyses, one study assessing the outcomes and risks of G-CSF in patients with CAD has reported an increased frequency of adverse outcomes (i.e., acute coronary syndrome) [64].

In conclusion, despite the promising results of small clinical trials or animal models using biochemical concepts to therapeutically promote the coronary collateral circulation, none of the approaches evaluated so far could be successfully translated into clinical practice. Besides the above mentioned limitations such as inefficient collateral formation (i.e., angiogenesis) or potentially harmful propagation of atherogenesis (i.e., the “Janus Phenomenon” [57]), a number of additional unresolved issues remains. These include questions relating to the dosage, the application route and the timing of administration of growth factors [44]. Importantly, considering that “no-option” patients with extensive CAD are the most likely candidates for coronary arteriogenesis, safety of any collateral-promoting substance is crucial [59].

### 4.2. Biophysical Concepts

The biophysical concept of arteriogenesis is to increase tangential vascular shear stress in preformed coronary anastomoses. One of the natural ways of increasing vascular shear stress is physical exercise [76]. However, because of different comorbidities it is often not feasible for patients with CAD to perform physical exercise training sufficiently. Thus, several other biophysical approaches have been introduced and will be described subsequently.

#### 4.2.1. Physical Exercise

The positive effects of physical exercise on the cardiovascular system have been known for a long time [77]. For instance, it was concluded in 1958 by Morris and Crawford that physically active people are less prone to develop stable CAD in comparison with sedentary people [78]. Physical exercise has a positive effect on several cardiovascular aspects such as vascular remodelling, increase of the maximal coronary blood flow (i.e., coronary flow reserve; CFR) as well as a decrease of coronary artherogenesis [79,80].

Concerning the effect of training on coronary collateral function, Scheel et al. observed an arteriogenetic effect of physical exercise in dogs with a constricted coronary artery whereas this effect was not observed in dogs without coronary stenosis [81]. In the groups with artificial coronary occlusion, exercise stress doubled the collateral growth and hence, the coronary flow reserve when compared to none exercised dogs. In humans, Zbinden et al. documented an increase of the quantitative parameter CFI in a proof-of-concept study [82]. They evaluated CFI, CFR and other cardiac parameters before and immediately after exercise training of a healthy marathon runner and demonstrated an increase of CFI from 0.23 to 0.37. Two small, non-randomized clinical trials have supported the positive effect of physical exercise on coronary collateral function, the increase in coronary cross-sectional area [83] as well as dose-dependent relation between training and increase in CFI [84]. Those results are in agreement with several other studies [80,85,86,87,88,89], which showed augmented perfusion by collateral vessels in response to exercise training. Besides, there have been other clinical trials failing to show a beneficial effect on the collateral circulation by exercise as assessed by angiographic imaging, but not by functional measurements [90]. However, the authors mention the limited validity of the angiographic approach and despite the negative outcome on coronary collateral formation, the exercise group had a significant better clinical outcome concerning the frequency of cardiac symptoms and the physical performance.

Recently, the first randomized clinical trial on the effect of physical exercise on coronary collateral function has been published [91]. Möbius-Winkler et al. randomly assigned 60 patients to two training groups (moderate- and high-intensity exercise with 10 h of training per week in each group) and one control group (usual care with encouragement to perform regular physical activity according to current recommendations). After four weeks, both exercise groups showed a significant increase in CFI (from 0.142 to 0.198, *p* = 0.005 respectively from 0.143 to 0.202, *p* = 0.004) without a statistically relevant difference between the training modalities whereas CFI in the control group remained unchanged (from 0.149 to 0.150, *p* = n.s.).

In conclusion, the positive effect of physical exercise on the human coronary collateral circulation has been repeatedly demonstrated. However, there remain important questions concerning the type and extent of physical exercise for optimal promotion as well as the implementation for patients with limited physical possibilities.

#### 4.2.2. External Counterpulsation (ECP)

“External counterpulsation therapy was first developed as a resuscitative tool to support the failing heart and was based on the hemodynamic principles of the intra-aortic balloon pump”, which is the augmentation of diastolic blood flow with consecutive improvement of coronary perfusion as well as ventricular afterload reduction [92]. ECP uses three pairs of pneumatic cuffs wrapped around each of the lower extremities. Those cuffs are sequentially inflated from distal to proximal triggered by the ECG. Besides augmenting diastolic blood flow and reducing ventricular afterload, ECP increases tangential endothelial shear stress triggering arteriogenesis. Used as a safe, effective and low-cost second line treatment in refractory angina pectoris, ECP has been shown to be efficacious in reducing CAD symptoms as well as improving exercise time [92,93,94,95,96]. Of note, the positive effect appears to outlast the actual, conventional seven week period of treatment [97].

The effects of ECP on the coronary collateral circulation have been evaluated in two invasive clinical trials. Buschmann et al. demonstrated in a non-randomized study a significant increase in CFI. Other invasive parameters obtained in that study as the index of microvascular resistance (IMR) or quantitative coronary angiography (QCA) remained unchanged and hence, the increase of CFI reflected a “true” improvement of the myocardial blood flow [98]. These results have been confirmed in a randomized, sham-controlled clinical trial with an increase in CFI from 0.125 to 0.174 at a four-week follow-up exam (*p* = 0.006) in the experimental, but not in the placebo group (CFI changed from 0.129 to 0.111, *p* = 0.14) [46].

Recently, the principle of external counterpulsation has been individualized in order to alleviate the side effects of ECP (i.e., the cumbersome procedure with high pressure levels), thus increasing its acceptance. The so called individual shear rate therapy (ISRT) adjusts the used treatment pressures of the pneumatic cuffs according to individually adapted intra-arterial shear rates to achieve the same effect with reduced pressure values [99]. The calculation is based on Doppler-flow parameters in the common carotid artery at different treatment pressure values. Due to this procedure, the individually calculated treatment pressure ranged between 160 to 220 mmHg instead of the regular treatment pressure of 250 to 300 mmHg.

#### 4.2.3. Coronary Sinus Reducer

The biophysical concept of the coronary sinus reducer is based on a perioperative approach during heart surgery with artificially narrowed coronary sinus for augmented retro-perfusion [100]. The exact pathophysiologic principle for a beneficial effect remains unclear [101]. One proposed mechanism assumes that the venous back pressure as applied in the coronary sinus is regionally balanced in the venous, but not in the vascular bed upstream of the microcirculation [102]. Based on the two regionally counteracting responses of the microcirculation during myocardial ischemia, that is maximal vasodilatation and increased myocardial compressive forces (i.e., augmented ventricular wall stress due to diminished myocardial thickness), regional imbalance in microvascular resistance with higher resistance in the ischemic area arises. Thus, augmented venous back pressure is able to reach the non-ischaemic microcirculation more easily than the ischaemic one, thereby increasing the microcirculatory resistance in the non-ischaemic zone. This leads to a flow diversion of arterialised blood to the ischaemic area at risk under the necessary condition of functional collateral connections originating from the non-ischaemic area [102].

Due to advances in percutaneous coronary intervention, and at the same time, increasing number of patients with refractory angina pectoris, several investigators have picked up this approach by using balloon-expandable, hourglass-shaped devices to physically narrow the coronary sinus [103,104,105]. In a first-in-man study, this device has demonstrated relief of angina pectoris in 12 out of 14 patients without options for coronary revascularization [103]. Subsequently, Verheye et al. performed a randomized, sham-controlled clinical trial in 104 patients, which confirmed the results of the first study [101]. They showed an improvement in Canadian Cardiovascular Society (CCS) score as well as quality of life. Nevertheless, exercise time and mean change in the wall motion index as assessed by means of dobutamine stress echocardiography remained unchanged [101]. Subsequently and to evaluate the role of this device in future clinical practice, a post marketing study is currently enrolling selected patients without revascularization options (called Reducer-I-study; NCT02710435). They plan to recruit over 400 patients and assess the clinical efficacy as well as the long-term outcome with follow-ups up to five years after implantation [104].

#### 4.2.4. Pharmacologic Biophysical Arteriogenesis

Ivabradine, a specific inhibitor of the I_f_-channel mainly expressed in sinoatrial nodal cells [106], specifically decreases the heart rate without affecting cardiac contractility, afterload or vasomotion as it occurs with beta-blockers [107,108]. Based on the biophysical rationale of diastolic prolongation by ivabradine with extension of diastolic vascular shear stress, a small randomized placebo-controlled trial has demonstrated a significant increase in CFI by ivabradine (from 0.107 to 0.152, *p* = 0.0461) [109]. This result is in accordance with several other trials, which have shown an arteriogenic effect on coronary arteries by initiating bradycardia [110,111,112]). However, despite the promotion of coronary collateral supply, ivabradine has not been shown to be efficacious with respect to cardiovascular outcomes (composite of death from cardiovascular causes or nonfatal myocardial infarction) in patients with stable CAD [106] in bigger randomized trials such as the BEAUTIFUL [113] and SIGNIFY trials [114].

## 5. Therapeutic Promotion of Extracardiac Coronary Supply

The anatomical connection between the IMAs and the coronary arteries via their most proximal branch (i.e., the pericardiacophrenic artery departing at the first or second intercostal space) is well documented [21,22,25] (Figure 1 and Figure 2). Additionally, due to the connection of the IMAs with the iliac external arteries via the superior and inferior epigastric arteries, collateral supply from the caudal side amounts to approximately two thirds of the flow during IMA patency [102]. This dual blood supply along with the direct anatomical connection to the coronary circulation provided the rationale for the IMA ligation method as a surgical treatment for angina pectoris. Using a small incision between the second and third rib under local anesthesia, transthoracic surgical access and ligation of the IMAs was first performed by D. Fieschi in 1939 (i.e., before the advent of modern cardiac surgery with cardioplegia and heart-lung-bypass) [115]. Later on, the approach was tested by a series of trials carried out in the late 1950s [116,117,118,119,120,121,122]. The primary end point of those clinical trials was angina pectoris and, inconsistently, ECG signs of myocardial ischemia. Battezzati et al. [116], after identifying anew a connection between both IMAs and the myocardium, reported consistent improvement in terms of cardiac symptoms in a uncontrolled trial among 304 CAD patients in 1959. Notable, this improvement was sustained during a follow-up of up to four years after the surgical intervention. In a further uncontrolled trial among 50 CAD patients, Kitchell et al. [123] reported similarly favorable results with symptomatic relief in 68% of the patients undergoing bilateral IMA ligation.

The following sham-controlled trials of bilateral IMA ligation in 35 CAD patients coined the phrase “surgery as placebo” in the context of their negative results [124,125,126]. Although the introduction of a sham-control study design in the context of surgical trials was seminal [127], the conclusion drawn from the negative results of the IMA ligation trials at hand is questionable. In the trial by Cobb et al. [125], angina pectoris relief was found in five of eight patients (63%) after IMA ligation and in five of seven patients (71%) after IMA sham ligation. Dimond et al. [126] reported nine of 13 patients in the verum and five of five patients in the sham-operation group, respectively. Thus, the abrupt stop of bilateral transthoracic IMA ligation was mainly caused by the advent of modern cardiac surgery with bypass grafting rather than by the slim evidence against IMA ligation claimed by the controlled trials. Especially because the soft study end point of angina pectoris would have required patient numbers at one order of magnitude higher than those recruited for the sham-controlled IMA ligation trials [125,126].

Because of the slim evidence against IMA ligation and the promising surgical results in terms of symptomatic relief, this therapeutic concept was revived 75 years after the first attempt using percutaneous interventional techniques. In the context of soft study endpoints, the first observational interventional study on the function of coronary supply by the IMAs has predefined intracoronary ECG (i.c.ECG) ST-segment elevation during coronary occlusion, not angina pectoris, as the first end point for ischemia [128]. In this trial, myocardial ischemia has been induced twice with and without simultaneous IMA occlusion by proximal coronary balloon occlusion in the process of CFI measurement. Further, to eliminate the effect of coronary collateral recruitment or ischemic preconditioning occurring during the second (but not the first) occlusion on the collateral circulation, CFI measurement with simultaneous IMA occlusion was performed before the control measurement without IMA occlusion. Despite this conservative study design, the approach showed a consistently reduced i.c.ECG ST-segment elevation during ipsilateral IMA with RCA or LAD occlusion as an expression of reduced ischemia. Further, CFI has been found higher in the presence versus the absence of IMA occlusion in 68% of the measurements, and overall, this difference amounted to +0.025 compared with the absence of IMA occlusion (*p* < 0.0001) [128]. However, contralateral IMA occlusion did not cause an effect indicating the necessity of anatomic vicinity. In this trial, functional connection between the coronary arteries and the IMAs was slightly less frequent in case of LAD with left IMA occlusion (25 of 30 measurements) than in the case of RCA with right IMA occlusion (28 of 30 measurements).

Based on those functional findings, an anti-ischemic therapeutic approach consisting in distal IMA occlusion by interventional techniques could be a promising therapeutic alternative to IMA bypass grafting. In an open-label proof-of-concept study, Stoller et al. investigated a catheter-based permanent IMA occlusion in the setting of the less frequently grafted right IMA among patients with ischemia in the RCA territory [129]. In this study, 50 patients with chronic stable CAD underwent permanent device occlusion of the distal right IMA. CFI of the RCA measured immediately before and six weeks after the IMA-occlusion showed a consistent increase from 0.071 at baseline to 0.132 (*p* < 0.0001). Further, this augmented coronary blood supply was reflected by the i.c.ECG as a direct measure of myocardial physiology revealing a decreased ischemia during RCA occlusion from baseline to follow-up examination (*p* = 0.0015). Figure 5 illustrates this increased collateral function along with decreased myocardial ischemia during coronary occlusion as outlined by an absent ST-deprivation in the ECG of the follow-up intervention. 

To conclude, augmentation of extracardiac coronary supply by permanent right IMA device occlusion is effective and feasible. However, if and how this increased collateral blood flow improves clinical outcome parameters is subject of current research. For this reason, a randomized, sham-controlled and double-blind clinical trial is currently enrolling patients (NCT03710070). It aims to include 250 patients in order to assess the clinical efficacy (measured as treadmill exercise time increment) in the next few years.

## 6. Conclusions

Based on the growing problem of patients with coronary artery disease and incomplete revascularization, several promising therapeutic alternatives for myocardial revascularization have been examined. Because of the known survival benefit of patients with a functional coronary collateral circulation, its promotion is a promising concept. However, until now, none of the evaluated concept could be implemented in daily clinical practice despite appealing results in clinical trials.

Biochemical concepts of angio- or arteriogenesis by growth factors seem to be prone to potentially harmful effects, since arteriogenesis shares many common mechanisms with inflammatory diseases, such as atherosclerosis. Thus, the risk benefit ratio is inappropriate and further research using growth factors was discontinued.

Biophysical concepts are based on increasing arteriogenesis via elevated tangential vascular fluid shear stress. Physical exercise training or external counterpulsation have been documented to positively affect clinical symptoms as well as coronary blood flow. The effect of both physical arteriogenic procedures is, however, transient (i.e., vanishes after its termination) and the time-consuming procedure of several hours per week limits the use to selected, highly motivated patients.

Alternative techniques such as coronary sinus reduction or promotion of extracardiac coronary supply by permanent occlusion of the distal internal mammary artery are promising approaches, since they have a permanent effect and ought to be efficacious in reducing myocardial ischemia to the effect that it is clinically relevant. Both approaches are being currently studied in ongoing clinical trials (NCT0271043 respectively NCT03710070) and the results of these investigations will clarify the clinical potential of those new therapeutic methods.

## Figures and Tables

**Figure 1 ijms-20-03726-f001:**
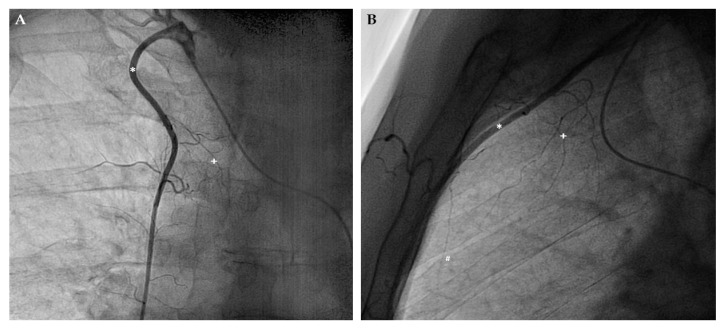
Angiographic demonstration of extracardiac coronary supply. (**A**) Posterior-anterior projection of the right internal mammary artery (IMA, marked by *) and its connection to the right coronary artery via the pericardiacophrenic branch (marked by +). (**B**) Lateral projection using the same markers. Noteworthy, additional branches of the IMA (marked by #) heading towards the heart.

**Figure 2 ijms-20-03726-f002:**
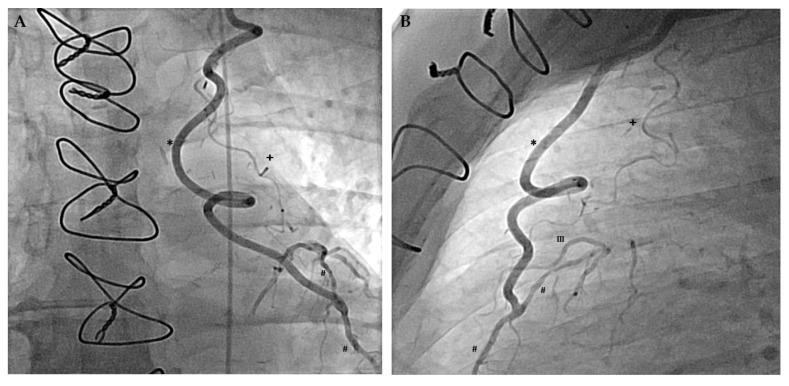
Angiographic demonstration of extracardiac coronary supply after coronary artery bypass surgery. (**A**) Posterior-anterior projection of the left internal mammary artery bypass (marked by a *) on the left anterior descending coronary artery (LAD, marked by a #). Upstream of the bypass anastomosis, retrograde filling of the LAD is incomplete revealing coronary occlusion, which triggered the arteriogenesis of the pericardiacophrenic branch (marked by a +) (**B**) Lateral projection using the same markers revealing the connection of the pericardiacophrenic branch with the third diagonal branch (marked by III).

**Figure 3 ijms-20-03726-f003:**
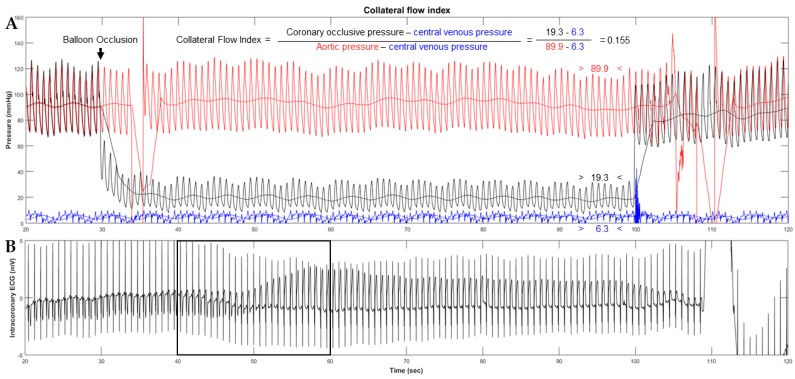
Collateral flow index (CFI) measurement. (**A**) Simultaneous recordings of mean and phasic aortic (red signals, Pao), coronary occlusive (black signals, Poccl) and central venous pressure (blue signals, CVP) immediately before (left side) and during coronary artery occlusion in a patient with poorly functional collaterals. (**B**) Detection of myocardial ischemia during the coronary artery occlusion by the intracoronary electrocardiogram (i.c.ECG). Immediately after balloon occlusion, the i.c.ECG shows marked electrical alternations with flipped T-waves and ST-segment elevation (marked by the black square). Generally, a CFI of >0.20–0.25 is related to absent signs of ischemia on i.c.ECG during a 1-min proximal coronary occlusion.

**Figure 4 ijms-20-03726-f004:**
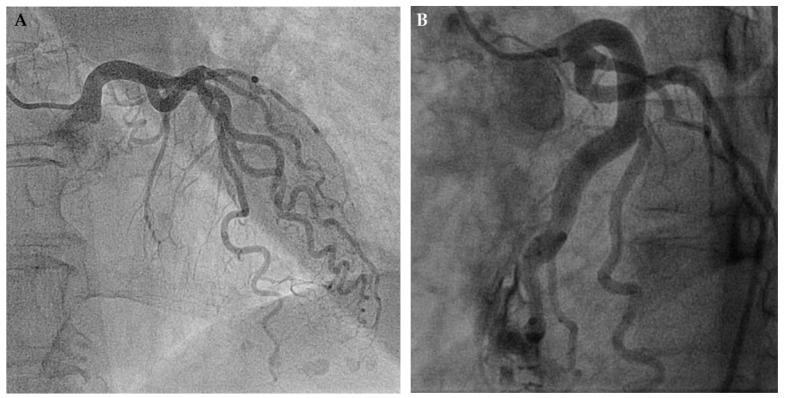
Angiographic presentation of two different pathophysiological etiologies of arteriogenesis. (**A**) Arteriogenesis in the course of hypertensive heart disease with concentric left ventricular hypertrophy. Enlarged myocardial mass is the driving force behind this arterial growth. (**B**) Arteriogenesis solely initiated by constant elevation of fluid shear stress. Iatrogenic drainage of the left anterior descending artery (LAD) into the right ventricular cavity after myocardial biopsy significantly increased coronary blood flow and consequently vascular size.

**Figure 5 ijms-20-03726-f005:**
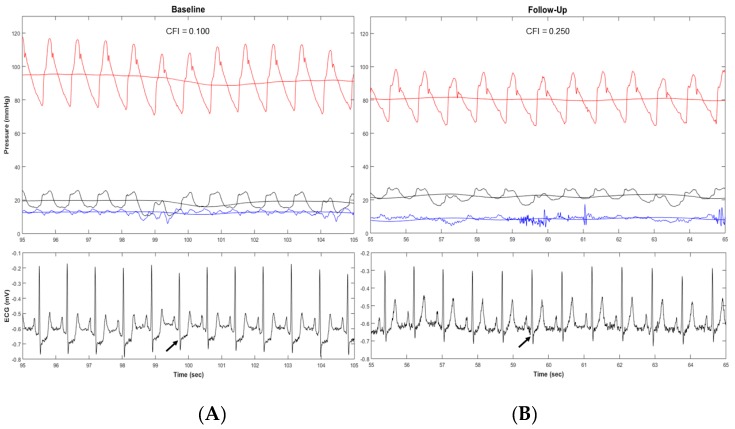
Collateral flow index (CFI) measurements of the right coronary artery (RCA) with corresponding electrocardiograms (ECG) after a one-minute proximal coronary balloon occlusion. (**A**) CFI measured immediately before permanent right internal mammary artery occlusion showing a collateral blood supply of 0.100 and marked ST-deprivations in the ECG as a sign of ischemia (marked with an arrow). (**B**) Six weeks after the permanent occlusion, CFI increased to 0.250 (+0.150). This augmented coronary blood supply is reflected by the ECG revealing a decreased ischemia without ST-deprivations (marked with an arrow).

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
