# Peer review of "The Human Coronary Collateral Circulation, Its Extracardiac Anastomoses and Their Therapeutic Promotion"

_ijms, 2019, doi:10.3390/ijms20153726_

Reviewer 1 Report

This is a review article on an emerging topic of coronary collateral circulation, an alternative to current standard of care to coronary artery disease. Mechanism and strategies of promoting coronary artery disease was discussed. The topic is novel and significant to clinical practice. 

However, the overall manuscript appears weak and poorly prepared. There is substantial text overlap with other publications (see attached cross-check file). The reviewed evidence is unbalanced (even somehow biased), and was far from sufficient to support most of the conclusions. The logic flow of the language was poor, perhaps due to significant text overlapping with other publications. 

One other major defect that severely compromised this review is that study results were presented as a laundry list, without providing critical synthesis, connection, or critique. For a review article, this doesn't give anything useful to its readers beyond a reasonably up-to-date bibliography.

Specific comment below: 

Please update the quotes from 2018 AHA using the 2019 statistics (Ref 1).

Fig 1. sinus node artery should be marked inn the images.

Title of 2.1 should simply be “coronary collateral circulation” in order to conform to the title of 2.2 

Fig 3. Spell out all abbreviations

Line 158, “small muscle cell” should be “smooth muscle cell”

Acknowledge the source of all figures.

Section 4.1.1 “Growth factors” is in fact the only “Biochemical concepts” discussed under Section 4.1. It does not appear necessary to isolate “Growth factors” as a stand-alone section.

It should be acknowledged that the evidence mentioned for “4.2.1 Physical exercise” were largely based on observational studies with extremely small sample size, and one of them was on healthy marathon runner. Such data was not representative of cardiovascular disease patients and whether these evidence mount to therapeutic relevance is questionable. Additional evidence supporting a role of physical exercise on coronary collateral function is much needed here, for instance, basic research in vitro or in animal models.

“Section 4.2.3 Coronary sinus reducer” contains large amount of text identical to web resources. The data presented here is so limited (ref 101) that it appears almost unlikely that such a strategy is significant.

The entire Section 5 is almost copied verbatim from a prior publication. There is no coherent logic from paragraph to paragraph and the sentences were haphazardly put together. The way data was presented in this section does not support the conclusion in the last paragraph.

The overall conclusion (Section 6) was poorly drafted. It should have highlighted the major findings of the main text, or should have pointed to future directions, or should have suggested practical recommendations based on balanced evaluation of all studies presented in the main text. In its current form, even removing the entire Section 6 wouldn’t cause too much of a difference to this manuscript. 

Reviewer 2 Report

Overall Considerations: 
In this manuscript Bigler Marius Reto et al. highlight the role of Human Coronary Collateral Circulation, Its Extracardiac Anastomoses and their Therapeutic Promotion

The manuscript is unique, innovative and alluring however I feel some specific issues need to be addressed before it reaches to a larger audience. 

Specific points:

Figure 1, 2, 3 and 5 need labelling (A, B, or C) and discussion of each labelled subpart. It is difficult for reader to follow authors narrative and corresponding figures.

Minor - Figure legends are whole inadequate and should be carefully rewritten to state succinctly what results are presented and what each picture element means. All figure legends would need to be extensively rewritten.

103- Replace is originating with originates

118- Replace Carioplegia with Cardioplegia

167- And underline take off

193 – Summarizes

208 – Expand MCP-1 

276- Replace 198 with 0.198

341- If channel Replace with subscript -  If  channel

# Weakness: In some places, the manuscript was a little difficult to read, so the Authors could have done a better job of paying attention to sentence structure and details. Manuscript requires an overall check of language for grammatical errors, punctuation and for sentences that either end abruptly or appear outside the flow. 

Author Response

Round  2

Reviewer 1 Report

The revised manuscript showed significant improvement. The authors have sufficiently addressed most of my concerns. The revised manuscript therefor meets the publication criteria in my opinion.